# From crisis to responsibility: The role of industry type, leadership style, and regulatory environment in shaping post-COVID-19 CSR initiative

**Yongming Zhu**[1], **Saima Kiran**[1]*, **Muhammad Salman**[1], **Shahid Sherwani**[1], **Faisal Sajjad**[2], **Naeem Ud Din**[1]

**1** School of Management, Zhengzhou University, Henan, China, **2** School of Economics and Management North China Electric Power University Changping District, Beijing, China

\* saim1234779@gmail.com

**Data Availability Statement:** All the relevant data is in manuscript and Supporting Information.

## Abstract

Corporate social responsibility (CSR) is a major concern in modern industries. Chinese industries are growing rapidly and delivering products and services to the market. The Covid-19 pandemic has changed the working style of every type of industry. The objective of this research was to determine the influence of leadership style and industry type on the regulatory environment. This research also aims to determine the impact of the regulatory environment on CSR from the perspective of Chinese industries. Data based on a sample size of 599 was used for data analysis, and Smart PLS 3.0 was used for the results of measurement model assessment and structural model assessment. This study highlighted that industry type and leadership style have a significant positive impact on the regulatory environment and CSR. The framework of this research is based on the identified research gap, and the findings of this study are significant for Chinese policymakers. Furthermore, the research also asserted practical implications that are reliable to advance practices in the regulatory environment and achieve CSR by Chinese firms. This study has several limitations that are required to be significantly addressed for the sustainability of organizations.

## 1. Introduction

In this era of rapid industrialization, social goals are required to be achieved by different kinds of industries. The responsibility of industrial organizations either in manufacturing or in the service sector, is required to improve practices for organizational sustainability [1]. Sustainable development is required to be improved for the safety of the environment. The stability of the environment is necessary for modern organizations that are required to be improved over time [2]. The access of industry to environmental regulations can be a helpful strategy to increase organizational performance. The concern of the industries should be to develop strategies that would be effective for the improvement of the environment [3]. When there is no sustainability in the environment, the industrial sector must work on it [4]. After the outbreak

**Funding:** The authors received no specific funding for this work.

**Competing interests:** The authors have declared that no competing interests exist.

of the Covid-19 pandemic, the situation changed altogether regarding industrial performance, and less attention was paid to the goals of corporations regarding the sustainability of the environment.

Corporate social responsibility (CSR) is considered one of the important factors influencing organizational work and performance [5]. Organizations that consider themselves responsible for achieving their organizational goals are fairly working to achieve sustainability [6]. Meanwhile, the functioning of any organization in environmental terms would be a way forward to achieving sustainability. China is the largest country from the perspective of primary, secondary, and tertiary industries [7]. The workings of this industrial sector required advanced work to improve organizational performance. Corporate level work in any country requires extra responsibilities that should be managed appropriately for the people [8]. The reliability of organizational work for environmental concerns can be a way forward to achieve stability in the environment. Industrial work after covid-19 changed traditional practices, which could be a negative way to advance organizational performance [9].

Hao, Xiao [10] highlighted that it is the responsibility of manufacturing industry management to protect the environment. The manufacturing units are required to follow the strict guidelines that are necessary for the stability of the environment. Furthermore, Hsiao, Jiang [11] reported that the advanced working of the organization sector and improved performance of employees with efficiency can be a way forward to achieve sustainability in the environment. Many firms are working to advance their performance to achieve organizational sustainability which is necessary for the productive performance of organizations [12, 13]. Firm performance for environmental sustainability depends on different factors, and organizational sustainability is also one of these factors [14, 15]. Access to information shared for public benefit should be improved, and people must be provided with the opportunity to increase their organizational behavior. Broadstock, Chan [16] highlighted that the environmental policies should be implemented strictly for those organizations that are contributing negatively to the environmental improvements.

However, the available studies in the literature do not explain the relationship of CSR with industry type and leadership style, which is critical to determine [17–19]. To address this loop in the body of knowledge, this research is conducted to test the influence of leadership style and industry type on the regulatory environment. This research was also initiated to investigate the impact of the regulatory environment on CSR from the perspective of Chinese industries. The framework of this research is based on the research gap, and the novel findings of this study are significant addictions to knowledge. This research has presented remarkable theoretical implications for the body of knowledge. Furthermore, the research also asserted practical implications that are reliable for advancing practices for the regulatory environment and achieving CSR by Chinese industries. This study has several limitations that are required to be addressed significantly for future directions to improve the body of knowledge.

## 2. Review of literature

### 2.1 Stakeholder theory

The stakeholder theory is a significant theory that discusses the relationships between employees, firms, and communities. According to this theory, the stakeholders' concerns should be focused on by the firms [20]. This theory highlighted that when the stakeholders' concerns are considered by the firms, better relationships should be developed [21]. This theory demonstrates that the stakeholders are concerned with a special kind of responsibility that is necessary to improve business practices. Indeed, environmental protection is also the responsibility of businesses, and better opportunities should be developed for the people for the protection of

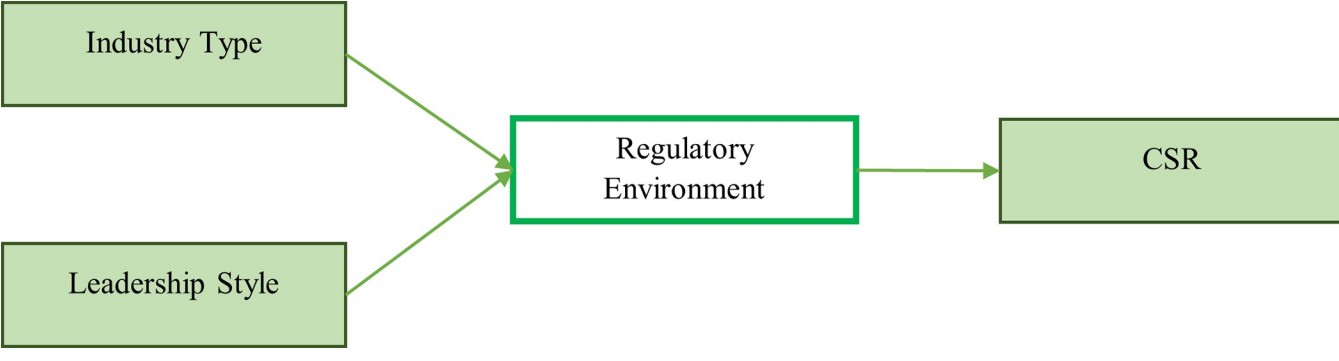

**Fig 1. Theoretical model of research.**

the environment. Based on the conceptual determination of this theory, the research model of the current study is also underpinned by this theory. This study has considered industry type and leadership style as the influencing components that are necessary for the regulatory environment. In this way, this research has considered the relationship between variables based on the theoretical underpinning of this research. Furthermore, this study has conceptually linked the stakeholder's theory's underpinned relationships with CSR. In this way, the framework of this research is developed (see Fig 1), highlighting industry type and leadership style as independent variables, CSR as the dependent variable, and the regulatory environment as a mediating variable.

## 2.2 Hypotheses development

The responsibility of every industry is to protect the environment. When any industry is working to improve its practices, the sustainability of the environment is necessary [3]. The regulation of the environment is necessary for environmental sustainability in modern times. Furthermore, the success of environmental sustainability is possible when the management of any organization is devoted to working for it [6]. Furthermore, the regulation of the government for environmental safety is also appropriate to ensure the working of organizations for the advancement of the environment [22]. Reasonable actions for environmental sustainability are required to be taken over time for its better benefit and advanced working [11]. The reliability of environmental concerns is possible when the management of any organization is devotedly working to improve the environment. The success rate of organizational performance and its advancement can be a way forward to improve environmental sustainability [16]. Awareness of the environment can improve employees' working behavior for better achievement of environmental goals in any industry. Industrial goals are achieved when environmental sustainability is to be achieved by the work of any organization [23]. Modern firms are largely concerned to improve environmental issues by working on realistic strategies to advance organizational performance.

Hypothesis 1: Industry type has a positive impact on the regulatory environment.

The role of leadership is critical to advancing organizational performance. The awareness of employees and leadership regarding environmental goals is the way forward to achieving environmental goals [8]. The safety of the environment for any firm is possible when the people are highly motivated to work for it. The top management strategies developed for the achievement of organizational goals in the best way are necessary to improve organizational performance [10]. Furthermore, those organizations that are not working appropriately to achieve their organizational goals are required to improve their performance strategically. The

actual working of any organization to advance environmental practices is possible when reasonable actions are taken by the management [24]. The role of management is critical to organizational sustainability, and the achievement of its goals. When the management of any organization has clear goals of environmental sustainability, it becomes easy for the employees to follow the management to achieve organizational performance [25]. The success rate of organizational work is possible through organizational achievement. The reliability of organizational work and its advanced work for the achievement of organizational goals can be a better strategy for the advancement of the environment. Many leaders are influencers of the employees to motivate them for better organizational performance.

Hypothesis 2: Leadership style has a positive impact on the regulatory environment.

CSR is necessary for any organization to achieve organizational sustainability and advancement in the environment [26]. Environmental concerns are achieved when the goals of any organization are realistic to achieve sustainability. A better working approach to organizational performance for environmental upgrading is necessary to advance the environment [27]. The reliability of environmental concerns and awareness of management for organizational goals is the way forward to achieving sustainability. However, when the organizational goals are to advance the environment, the performance of management and organizational practices are improved to achieve these goals [28]. The success rate of industry performance for the implementation of CSR is directly dependent on organizational goals. When the goals of any organization are in the right direction, awareness of organizational sustainability is achieved [29]. Improvement in the performance of management is key to achieving organizational goals. The success rate of the organization working for environmental awareness among the employees can be a better strategy to achieve organizational sustainability. Modern industries are required to work in the environmental sustainability direction to achieve the goals of sustainability [30]. Furthermore, CSR goals are a positive way to achieve sustainability in the environment.

Hypothesis 3: Regulatory environment has a positive impact on corporate social responsibility.

Organizational performance is necessary for the direction of environmental sustainability [2]. Regulation to protect the environment by any government are necessary to be implemented. The reliability of government regulations for organizational performance has increased over time. The implementation of government regulations for achievement of the sustainability goals is required to be managed by organizational performance [31]. The approach of any industry to working for the sustainability of the environment is necessary. Many organizations are working well to advance their practices for organizational sustainability. The role of every type of industry is to work on the policies that are necessary for organizational sustainability [32, 33]. The improvement in organizational goals and the worth of organizational work for environmental protection can motivate employees to work in the right direction for the achievement of sustainability. However, many organizations are working fairly to achieve the goals of sustainability, and these organizations must protect the environment [34]. The stable working of top management and long-term policies for organizational sustainability is the way forward to get a better way of organizational advancement. The reliable performance of employees and management of any industry could be a way forward to achieve the goals of CSR.

Hypothesis 4: Regulatory environment has positive mediation between industry type and corporate social responsibility.

The working style of leadership in any industry can influence organizational performance, both positively and negatively [1]. Those organizations that have developed appropriate policies to achieve sustainability are required to work in a better direction to achieve organizational goals. The reliability of organizational performance improves or decreases over time.

The organizational management should develop appropriate policies for the safety of the environment, and these policies should be implemented to get a better direction toward organizational goals [7]. The awareness of management for the achievement of organizational goals is the way forward to success in organizational performance. The stable working of an organization for the sustainability of the environment is the way forward to achieving organizational goals for environmental sustainability [12]. The regulations of the government used for advancement of organizational work can become better strategies to improve organizational performance productively. The better way to improve organizational work can be a factor in advancing the sustainability of the environment [35]. Many firms are working in the right direction to achieve the organizational goals of sustainability, and the performance of these organizations matters a lot in environmental upgradation.

Hypothesis 5: Regulatory environment has positive mediation between leadership style and corporate social responsibility.

## 3. Methodology

This research has considered reflective measurement item to collect the data. Therefore, a comprehensive process of measurement items development is used in this research. Based on this process, the literature was reviewed at the first stage. Secondly, the operationalization of the study's variables was made. These variables were considered appropriate to determine its findings. A pool of scale items was created with the help of experts, and the face validity of the measurement items was determined. These scale items were developed based on each dimension of the variables to be measured. However, to determine the validity and reliability of the items, the data was collected from a few respondents in the research population. The population of this research was based on the management of the Chinese industry. Therefore, data from only 80 respondents was collected for the preliminary tests of research items' reliability and validity. The data was collected after introducing the objective of the research to the respondents. They were asked to provide appropriate data because the analysis would be used for the reliability and validity tests of the questionnaire. This collected data was analyzed with AMOS 26, and the findings of exploratory factor analysis and confirmatory factor analysis were determined. The results of exploratory factor analysis reported that the research data has reliability and validity, but the insignificant measurement items with low levels of validity were removed at this stage. Furthermore, the findings of confirmatory factor analysis reported that an appropriate correlation is achieved for research data between the variables. Thus, the developed items for this research are considered appropriate for the final data collection.

The population of this research was the management level employees of the Chinese financial service industry, and the respondents from banks and insurance companies were targeted to collect data. To collect the data, a convenience sampling approach was used. This approach is best when the population is large and unknown [36]. Similarly, various recent studies have used this approach for data collection [37, 38]. A total of 650 questionnaires were surveyed during the month of May 2023. A brief introduction to the research was also given to the respondents, and informed verbal consent was obtained. The respondents were asked to provide appropriate data for this research. All of their quarries were addressed on time, and 599 questionnaires were collected back. This research has used Smart PLS 3.0 for final data analysis and findings. The results of the measurement model and structural model are used to determine the validity & reliability of the hypotheses and the path findings in this research. These findings are significantly used to determine the final relationships based on theoretical hypotheses with empirical support.

# 4. Data analysis and findings

The normality of this research data is tested at the initial stage. The normality test is used to determine the findings of any research to be considered for further analysis. It is necessary to determine the normality of data before utilizing the collected data for final analysis. The findings of missing values are determined at the initial stage to determine the normality. When there are no missing values in the data, then the possibility of data normality increases. Furthermore, the skewness and kurtosis values are considered to test the normality of research data. These values are tested by different methods, but this research has tested skewness and kurtosis findings by inserting the collected data into Smart PLS 3.0. The findings of kurtosis and skewness between -1 and +1 are significantly accepted for data normality [39]. These findings are reliable to consider, and the results of this research highlight that the study has achieved significant validity. The data for skewness, kurtosis, and missing values are reported in Table 1.

Furthermore, this research has used data analysis by the PLS Algorithm method to determine the reliability and validity of the measurement items at the individual level. For this purpose, the factor loadings are determined. Indeed, the factor loadings are used to determine whether the measurement items used for any variable are reliable or insignificant. The recommended threshold for factor loadings > 0.60 is achieved by the findings of this research [40]. If the recommended values are not achieved, the scale items are deleted and the data is analyzed again. However, the significant factor loadings values for this research are reported in Table 2. The findings are also displayed in Fig 2.

Table 1. Data normality.

| No. | Items | Missing | Mean | Median | Min | Max | Standard Deviation | Excess Kurtosis | Skewness |
|-----|-------|---------|-------|--------|-----|-----|--------------------|-----------------|----------|
| 1 | IT1 | 0 | 4.086 | 4 | 1 | 5 | 1.052 | 0.874 | -0.171 |
| 2 | IT2 | 0 | 3.579 | 4 | 1 | 5 | 1.134 | -0.256 | -0.580 |
| 3 | IT3 | 0 | 3.968 | 4 | 1 | 5 | 1.070 | 0.107 | -0.867 |
| 4 | IT4 | 0 | 3.932 | 4 | 1 | 5 | 1.038 | 0.152 | -0.827 |
| 5 | IT5 | 0 | 3.446 | 4 | 1 | 5 | 1.148 | -0.467 | -0.423 |
| 6 | IT6 | 0 | 3.586 | 4 | 1 | 5 | 1.134 | -0.479 | -0.495 |
| 7 | IT7 | 0 | 3.571 | 4 | 1 | 5 | 1.217 | -0.585 | -0.544 |
| 8 | LS1 | 0 | 4.432 | 5 | 1 | 5 | 0.976 | 0.803 | -0.836 |
| 9 | LS2 | 0 | 4.193 | 5 | 1 | 5 | 1.072 | 0.902 | -0.284 |
| 10 | LS3 | 0 | 3.968 | 4 | 1 | 5 | 1.196 | 0.325 | -0.096 |
| 11 | LS4 | 0 | 4.054 | 4 | 1 | 5 | 1.190 | 0.463 | -0.179 |
| 12 | LS5 | 0 | 3.904 | 4 | 1 | 5 | 1.169 | -0.010 | -0.918 |
| 13 | LS6 | 0 | 4.043 | 4 | 1 | 5 | 1.149 | 0.459 | -0.136 |
| 14 | RE1 | 0 | 4.104 | 4 | 1 | 5 | 1.099 | 0.875 | -0.246 |
| 15 | RE2 | 0 | 3.743 | 4 | 1 | 5 | 1.289 | -0.513 | -0.747 |
| 16 | RE3 | 0 | 3.654 | 4 | 1 | 5 | 1.212 | -0.505 | -0.629 |
| 17 | RE4 | 0 | 3.975 | 4 | 1 | 5 | 1.057 | 0.237 | -0.899 |
| 18 | RE5 | 0 | 4.032 | 4 | 1 | 5 | 1.015 | 0.454 | -0.970 |
| 19 | RE6 | 0 | 3.996 | 4 | 1 | 5 | 1.047 | 0.568 | -0.006 |
| 20 | RE7 | 0 | 4.029 | 4 | 1 | 5 | 1.069 | 0.546 | -0.063 |
| 21 | CSR1 | 0 | 3.943 | 4 | 1 | 5 | 1.136 | 0.296 | -0.991 |
| 22 | CSR2 | 0 | 3.764 | 4 | 1 | 5 | 1.138 | -0.062 | -0.785 |
| 23 | CSR3 | 0 | 3.846 | 4 | 1 | 5 | 1.141 | 0.125 | -0.900 |
| 24 | CSR4 | 0 | 4.021 | 4 | 1 | 5 | 1.183 | 0.496 | -0.161 |
| 25 | CSR5 | 0 | 3.714 | 4 | 1 | 5 | 1.104 | -0.220 | -0.633 |

**Table 2. Factor loadings.**

| Items | CSR | Industry Type | Leadership Style | Regulatory Environment |
|---|---|---|---|---|
| CSR1 | 0.904 | | | |
| CSR2 | 0.881 | | | |
| CSR3 | 0.916 | | | |
| CSR4 | 0.788 | | | |
| CSR5 | 0.674 | | | |
| IT1 | | 0.637 | | |
| IT2 | | 0.684 | | |
| IT3 | | 0.802 | | |
| IT4 | | 0.821 | | |
| IT5 | | 0.778 | | |
| IT6 | | 0.730 | | |
| IT7 | | 0.831 | | |
| LS1 | | | 0.656 | |
| LS2 | | | 0.875 | |
| LS3 | | | 0.849 | |
| LS4 | | | 0.874 | |
| LS5 | | | 0.853 | |
| LS6 | | | 0.849 | |
| RE1 | | | | 0.630 |
| RE2 | | | | 0.731 |
| RE3 | | | | 0.806 |
| RE4 | | | | 0.836 |
| RE5 | | | | 0.866 |
| RE6 | | | | 0.835 |
| RE7 | | | | 0.860 |

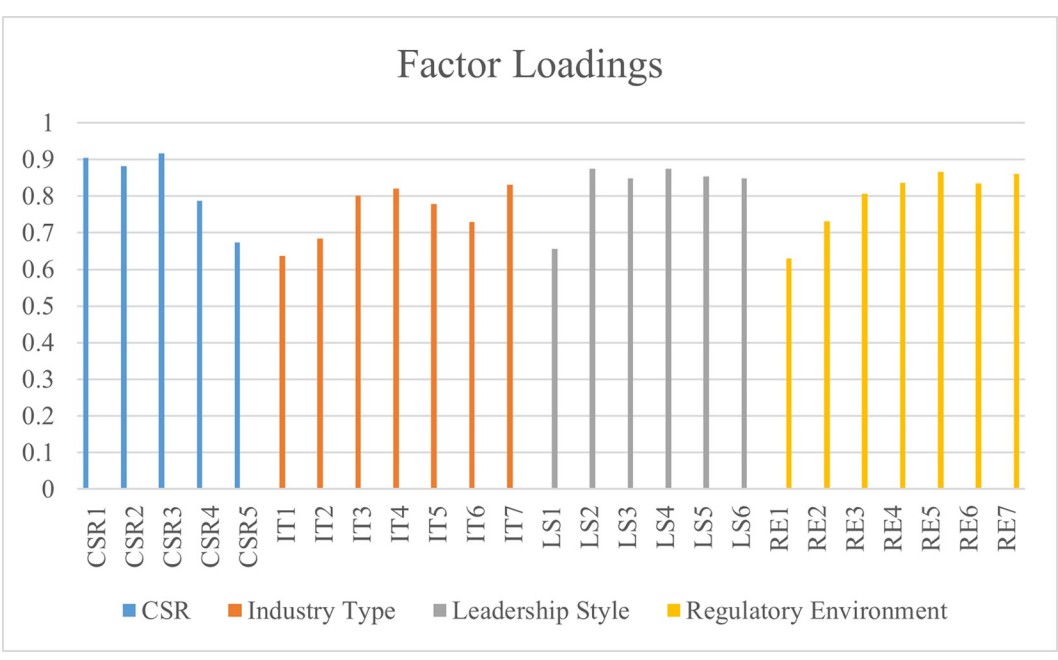

**Fig 2. Factor loadings.**

**Table 3. Cronbach alpha, composite reliability, and average variance extracted.**

| Variables | Cronbach's Alpha | Composite Reliability | Average Variance Extracted |
|---|---|---|---|
| CSR | 0.840 | 0.892 | 0.638 |
| Industry Type | 0.875 | 0.904 | 0.574 |
| Leadership Style | 0.908 | 0.929 | 0.688 |
| Regulatory Environment | 0.903 | 0.924 | 0.638 |

The findings of Cronbach alpha, composite reliability, and average variance extracted are determined in this research. These findings are used to significantly test the validity and reliability of research data. The average variance extracted findings are used to test the variable between research data, and the findings loaded more than 0.50 are significantly accepted for it [41]. Furthermore, the findings of composite reliability are used to test the reliability of research data, and the findings of more than 0.70 are accepted as significant [42]. Finally, the findings of Cronbach alpha are used to test the validity of the research data, and the findings loaded more than 0.70 are significantly accepted [43]. In this way, the data reported in Table 3 highlighted that the significant values for all three thresholds of the factors are achieved, and the data collected for this research has reliability and validity. These findings are also shown in Fig 3.

The discriminant validity of this research data is also tested because it is necessary to confirm that the data for this research is appropriate. The findings of discriminant validity are tested with Heteritrait-Monotrait (HTMT) method. This method is based on a second generation of data analysis. The findings in HTMT analysis should not be more than 0.90 for significant discriminant validity [44]. The analyzed data showed that the findings of this research have appropriate discriminant validity, as the data for this research has achieved significant HTMT. The results of HTMT are shown in Fig 4 and Table 4.

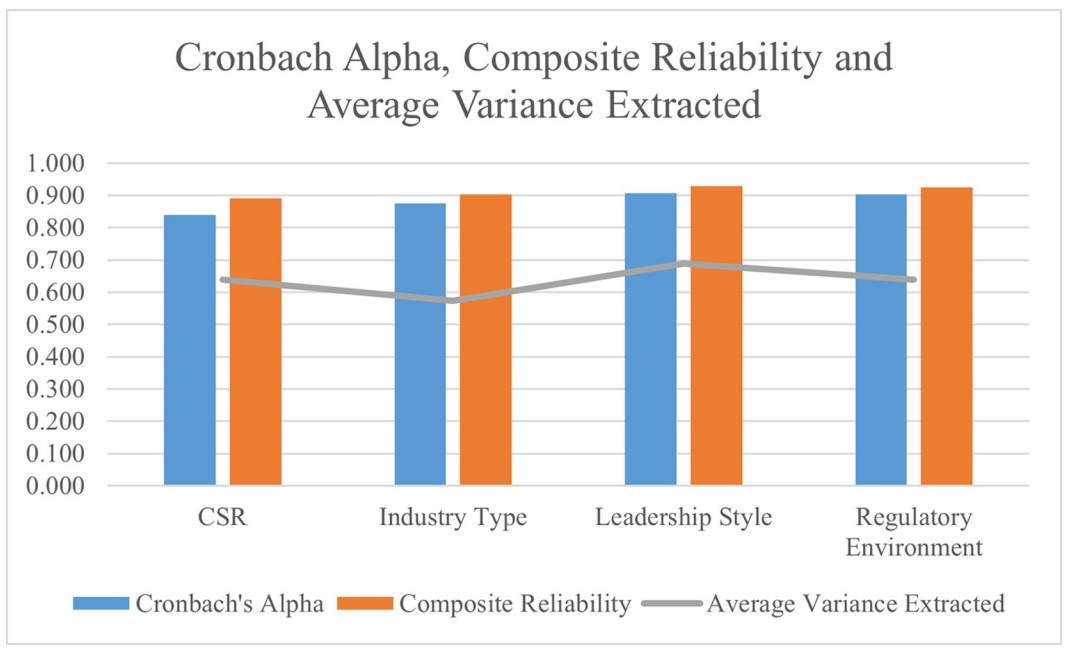

**Fig 3. Cronbach alpha, composite reliability, and average variance extracted.**

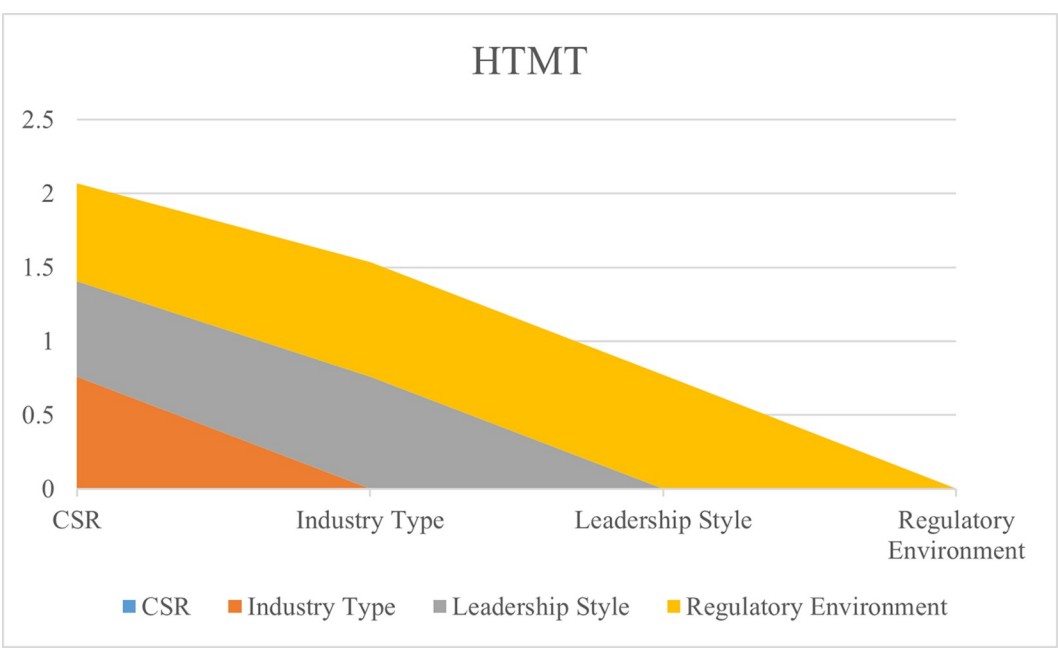

**Fig 4. HTMT.**

The approach of cross-loadings is also tested to determine the findings of discriminant validity. These findings are significantly used to determine the discrimination between the data for each measurement item. Simply, this method is used to test the discriminant validity at the measurement item level. However, the factor of cross-loadings is significant when the findings of one variable's measurement items are greater than the findings of other variables' measurement items that are correlated with it [45]. The results of cross-loadings highlighted that this research has achieved significant discriminant validity, and the data is shown in Table 5. The findings of cross-loadings are also displayed in Fig 5.

The outcomes of the structural model are used for hypotheses testing. The findings of t-statistics are used to determine the rejection and acceptance of relationships. The findings of hypothesis 1 reported that industry type has a positive and significant impact on the regulatory environment. Furthermore, the findings of this research reported that hypothesis 2 is accepted and leadership style has a positive and significant impact on the regulatory environment. Thirdly, the findings of hypothesis 3 reported that the regulatory environment has a positive and significant impact on corporate social responsibility. Fourthly, the findings of hypothesis 4 highlighted that the regulatory environment has positive and significant mediation between industry type and corporate social responsibility. Lastly, the outcomes of hypothesis 5 demonstrated that the regulatory environment has a positive and significant mediation between leadership style and corporate social responsibility. Hence, all five theoretically developed

**Table 4. HTMT.**

| Variables | CSR | Industry Type | Leadership Style | Regulatory Environment |
|---|---|---|---|---|
| CSR | | | | |
| Industry Type | 0.762 | | | |
| Leadership Style | 0.643 | 0.758 | | |
| Regulatory Environment | 0.665 | 0.776 | 0.773 | |

**Table 5. Cross loadings.**

| Items | CSR | Industry Type | Leadership Style | Regulatory Environment |
|---|---|---|---|---|
| CSR1 | **0.904** | 0.603 | 0.511 | 0.763 |
| CSR2 | **0.881** | 0.589 | 0.513 | 0.757 |
| CSR3 | **0.916** | 0.583 | 0.507 | 0.782 |
| CSR4 | **0.788** | 0.450 | 0.483 | 0.692 |
| CSR5 | **0.374** | 0.329 | 0.234 | 0.300 |
| IT1 | 0.348 | **0.637** | 0.551 | 0.435 |
| IT2 | 0.461 | **0.684** | 0.417 | 0.444 |
| IT3 | 0.501 | **0.802** | 0.584 | 0.571 |
| IT4 | 0.492 | **0.821** | 0.622 | 0.581 |
| IT5 | 0.528 | **0.778** | 0.453 | 0.552 |
| IT6 | 0.492 | **0.730** | 0.439 | 0.475 |
| IT7 | 0.598 | **0.831** | 0.532 | 0.585 |
| LS1 | 0.254 | 0.451 | **0.656** | 0.355 |
| LS2 | 0.425 | 0.542 | **0.875** | 0.562 |
| LS3 | 0.490 | 0.566 | **0.849** | 0.568 |
| LS4 | 0.499 | 0.532 | **0.874** | 0.608 |
| LS5 | 0.563 | 0.646 | **0.853** | 0.680 |
| LS6 | 0.563 | 0.618 | **0.849** | 0.672 |
| RE1 | 0.426 | 0.530 | 0.573 | **0.630** |
| RE2 | 0.607 | 0.637 | 0.729 | **0.731** |
| RE3 | 0.691 | 0.561 | 0.489 | **0.806** |
| RE4 | 0.730 | 0.477 | 0.445 | **0.836** |
| RE5 | 0.732 | 0.554 | 0.543 | **0.866** |
| RE6 | 0.749 | 0.516 | 0.485 | **0.835** |
| RE7 | 0.804 | 0.580 | 0.517 | **0.860** |

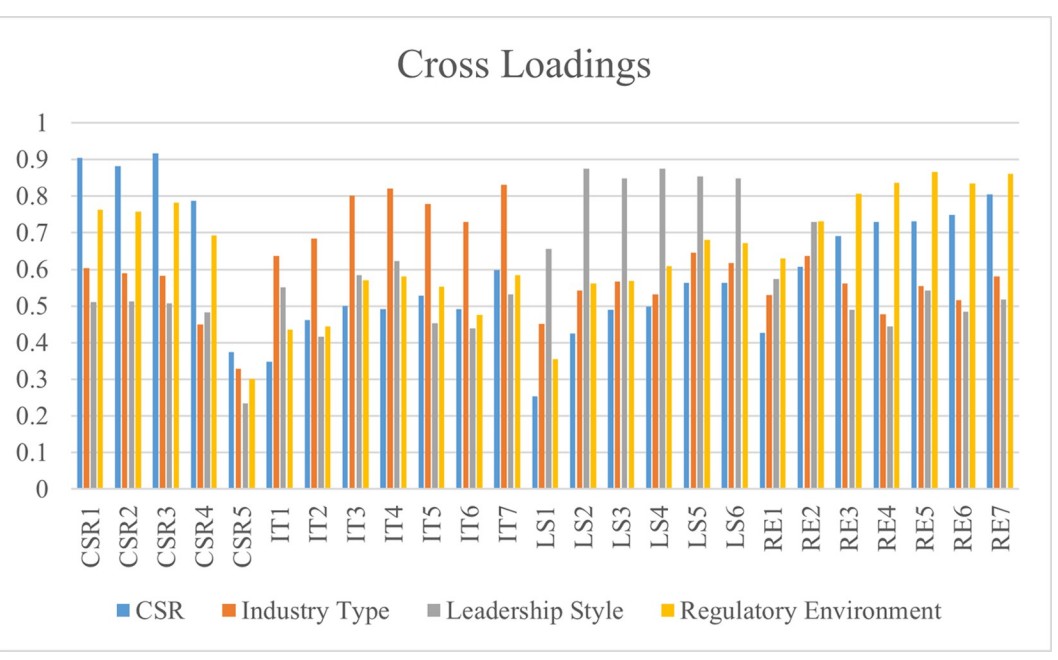

**Fig 5. Cross loadings.**

**Table 6. Hypotheses testing.**

| No | Hypotheses | Original Sample | Standard Deviation | T Statistics | P Values |
|---|---|---|---|---|---|
| 1 | Industry Type -> Regulatory Environment | 0.391 | 0.065 | 5.995 | 0.000 |
| 2 | Leadership Style -> Regulatory Environment | 0.443 | 0.063 | 7.057 | 0.000 |
| 3 | Regulatory Environment -> CSR | 0.856 | 0.020 | 42.617 | 0.000 |
| 4 | Industry Type -> Regulatory Environment -> CSR | 0.334 | 0.058 | 5.791 | 0.000 |
| 5 | Leadership Style -> Regulatory Environment -> CSR | 0.379 | 0.055 | 6.894 | 0.000 |

Significant: t > 1.96

relationships of this research are significantly accepted. The findings are presented in Table 6. However, the graphical representation of these findings is available in Fig 6.

## 5. Discussion and conclusion

The findings of this research are based on the data collected with the structural equation model. The hypotheses testing is done with t-statistics to determine the findings of this research. The empirical data for this research is considered appropriate to support the theoretically developed relationships. This research highlighted that industry type has a significant and positive impact on the regulatory environment. However, this relationship is new to our knowledge but has some theoretical support from the findings of existing studies in the literature. According to Sadiq, Nonthapot [8], the preservation of the environment is a duty shared by all sectors of society. The sustainability of the environment is essential when any industry strives to advance its methods. In the current day, environmental control is essential for the sustainability of the environment. According to Dongfang, Ponce [27], when the management of any organization is committed to working towards it, environmental sustainability can succeed. Additionally, the government's environmental safety regulations are appropriate to

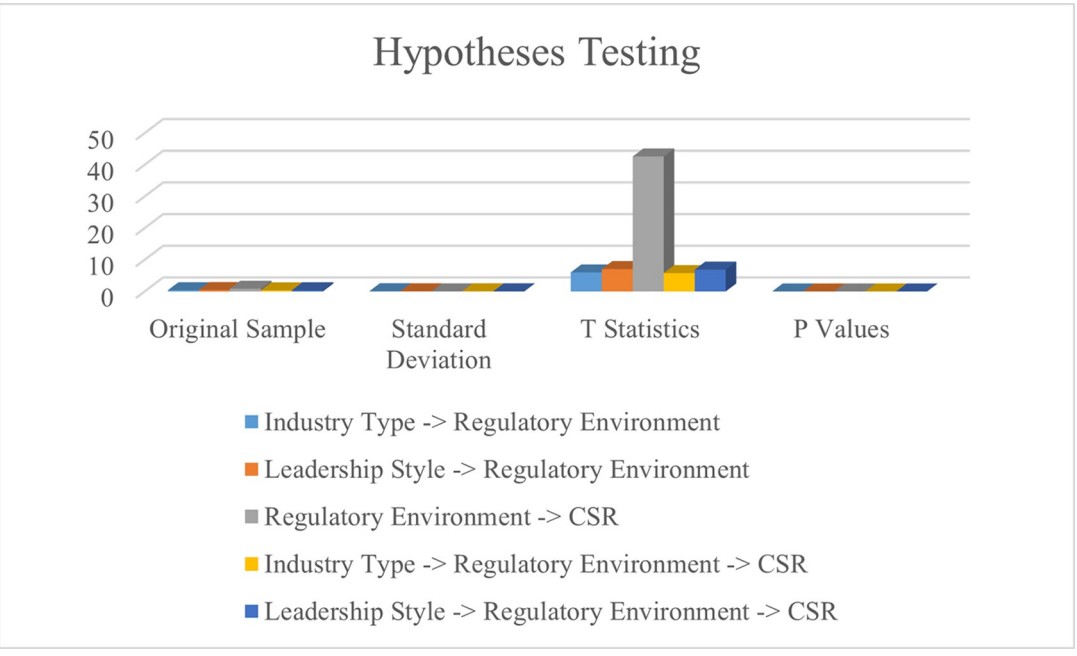

**Fig 6. Hypotheses testing.**

ensure the operation of organizations that promote environmental advancement. According to Hao, Xiao [10], for improved advantage and more advanced operation, suitable activities for environmental sustainability must be undertaken over time. Once the leadership of an enterprise is committed to enhancing its sustainability, its environmental concerns can be trusted. According to Poursoleyman, Mansourfar [28], to attain the objectives of ecological responsibility, modern industries must operate with the most environmentally friendly approaches. The CSR objectives are also a good way to promote environmental sustainability. To move in the direction of environmental sustainability, organizational performance is essential. According to Sun and Li [35], any government's environmental protection regulations must be put into practice. Government laws' dependability for measuring organizational performance has grown over time.

Secondly, this research highlighted that leadership style has a significant and positive impact on the regulatory environment. According to Xu, Chen [30], organizational performance must be used as a management tool to oversee the application of governmental rules for the accomplishment of sustainable goals. According to Waheed, Shehzad [25], any industry must have a proactive stance in support of environmental sustainability. Many organizations are making great strides in improving their organizational sustainability practices. According to Huang, Chen [26], the key to improving the efficiency of organizations is for the administration to be aware of the importance of achieving organizational goals. The best method to move forward in achieving organizational objectives for environmental sustainability is through steady organizational operations. According to Hu, Qu [23], government rules that are used to enhance organizational working can be improved upon to provide more effective tactics for enhancing organizational performance. The more effective means of enhancing organizational performance may contribute to greater environmental sustainability. According to Zhang, Lu [1], the performance of these organizations is very important for the improvement of the environment, and many businesses are working in the right direction to fulfill their organizational goals of sustainability. The key to improving the efficiency of organizations is for the administration to be aware of the importance of achieving organizational goals. According to Teng, Teng [2], the best method to move forward in achieving organizational objectives for environmental sustainability is through steady organizational operations. Government rules that are used to enhance organizational working can be improved upon to provide more effective tactics for enhancing organizational performance.

Thirdly, this research highlighted that the regulatory environment has a significant and positive impact on CSR. According to Liu, Qiao [46], the more effective means of enhancing organizational performance may contribute to greater environmental sustainability. According to Li, Hu [32], the performance of these organizations is very important in the improvement of the environment, and many businesses are working in the right direction to fulfill their organizational goals of sustainability. The rate of development of organizational performance can help advance ecological sustainability. According to Wei, Ayub [34], any industry can better achieve environmental goals by improving employee behavior through environmental knowledge. When any organization works to attain environmental sustainability, industrial aims are also achieved. According to Abbas, Wang [47], by developing practical measures to boost organizational performance, modern businesses are primarily concerned with addressing environmental challenges. The leadership position is essential to improving organizational performance. Accordingly, the way forward for achieving the environmental goals is for staff members and managers to be aware of them. The path forward to achieving sustainability is the dependability of environmental concerns and knowledge of management for organizational goals.

The fourth hypothesis findings highlighted that the mediating role of the regulatory environment has a significant influence on the relationship between industry type and CSR. This

relationship is in its infancy stage, but the empirical findings support it. The findings of existing studies in the literature also supported these relationships in the literature. According to Hsiao, Jiang [11], management effectiveness and organizational practices are enhanced to meet these objectives when the organization's goals are to advance the environment. The organizational objectives have a direct impact on the success rate of industrial performance for the implementation of CSR. According to Zhong, Sun [29], the understanding of organizational sustainability is attained when the aims of any organization are moving in the proper direction. The achievement of organizational goals depends on management performance improvement. According to Jaiyeoba, Hossain [31], a better approach to achieving organizational sustainability may be to increase employee environmental awareness through organizational efforts. Personnel can be inspired to work in the appropriate direction for the accomplishment of sustainability goals through improvements in organizational goals and the value of organizational work for environmental protection. According to Hsiao, Huang [7], a lot of organizations are making fair efforts to meet sustainability objectives, and these organizations need to safeguard their surroundings. The best path forward for organizational growth is through the steady operation of top management and long-term plans for organizational sustainability.

Finally, the findings of the last hypothesis highlighted that the mediating role of the regulatory environment has a significant influence on the relationship between leadership style and CSR. The findings of existing studies in the literature also supported these relationships. According to Freze, Korneev [3], any industry's management and employees putting in consistent work could be a step in the right direction for achieving CSR objectives. Each industry's leadership working style has the potential to both positively and negatively impact organizational performance. According to Zhong, Zhao [24], a lot of executives influence their followers to inspire them to perform better for the organizations. Any organization that wants to attain organizational sustainability and environmental growth must engage in CSR. When an organization's goals are achievable and sustainable, environmental concerns are addressed. According to Hao, Xiao [10], to improve organizational performance for environmental improvement, a better working methodology is required. Every industry has a responsibility to work on the regulations required for organizational sustainability. According to Broadstock, Chan [16], organizations that have created adequate policies for achieving sustainability are expected to operate more effectively towards achieving organizational objectives. Over time, the dependability of organizational performance either increases or decreases. According to Li, Zhong [6], any company can have a safe working environment if its employees are highly motivated to do so. A better method to attain environmental advancement is through the consistency of organizational functioning and its advanced effort for organizational aims. According to Zhang, Li [12], to improve the direction of organizational goals, organizational management should design appropriate policies for the safety of the environment and implement these policies.

In a nutshell, the loop in the literature is closed as the research investigated the empirical findings for the relationship between industry type, leadership style, and CSR. Furthermore, this research also closed a loop in the literature by confirming the mediating role of the regulatory environment between industry type, leadership style, and CSR. Therefore, the findings of this research are critical both theoretically and practically. The available data used in this research is free from any copyright issues. Finally, the findings of this research are generalized based on their findings. The study generalized that the type of industry and the style of leadership are critical factors that influence the leadership style and their working approach. Finally, this study also highlighted that the environment should be regulated for the significant implications of CSR practices.

## 6. Theoretical and practical implications

The findings of this research are important from theoretical as well as practical perspectives. This research has extended our knowledge in light of stakeholder theory. This research has introduced newly developed relationships in the literature. Firstly, this study has introduced in the literature that industry type has a positive impact on the regulatory environment, and this relationship was not reported in the literature before this research. Furthermore, this study added to the literature that the leadership style has a positive impact on the regulatory environment. In the same way, this relationship is newly introduced in the literature and not reported by the findings of existing studies. Thirdly, this research demonstrated in the literature that the regulatory environment has a positive impact on corporate social responsibility. Meanwhile, this relationship is also new in the literature and has not been reported in existing studies. This research furthermore highlighted that the regulatory environment has a positive mediation between industry type and corporate social responsibility. Finally, the study highlighted in the literature that the regulatory environment has a positive mediation between leadership style and corporate social responsibility. In this way, the relationships developed by this research are worthy of knowledge because new directions regarding corporate social responsibility under the light of stakeholder theory are highlighted by this research.

The findings of this research have practical implications, as they demonstrated that environmental regulations are possible with industry type and leadership style. In this way, the work of leadership and their style matter a lot regarding environmental issues. The success and failure of environmental regulations are based on the working practices of the leadership. These regulations are required to be improved over time to achieve corporate social responsibility. Firstly, the primary industry is required to work on the environmental improvement that is necessary to achieve sustainability in it. The primary industries are required to improve efficiency in the utilization of natural resources, which is a way forward to success in it. In the meantime, the secondary industry is also required to work to advance efficiency in the utilization of natural resources with the external factors that are critical to managing for better work. On the other hand, the industry of the tertiary sector is also required to be improved for the utilization of resources to protect the environment. In the meantime, access to natural resource protection for the safety of the environment is required to be improved by every type of industry. In this way, corporate social responsibility can be significantly increased to get better output.

## 7. Future directions

Based on the empirical results, this study highlighted that the impact of industry type and leadership style is significant on the regulatory environment and CSR. The study also introduced the regulatory environment as a mediator between leadership style, industry type, and CSR. Moreover, the framework of this research is based on the research gap, and the novel findings of this study are significant additions to the knowledge. The study has presented remarkable theoretical implications in the body of knowledge. On the other hand, the research also asserted practical implications that are reliable to advance the practices for the regulatory environment and achieving CSR by Chinese industries. This study has several limitations that are required to be addressed significantly for future directions to improve the body of knowledge. This study has limitations in its data collection because the data was collected only from the management of the industries in China. However, they understand the relationship between leadership style, industry type, and regulatory environment, so the data collected from the employees would be appropriate for the findings. In this way, this limitation of research must be addressed by future studies to contribute significant findings to the literature. In the

meantime, this research has not considered environmental awareness as a factor in the research framework. This exclusion of environmental awareness limits the findings of this research. In this way, future studies are required to consider environmental concern as moderating variable to determine the findings and contribute to the literature. This research has collected cross-sectional data for its findings, but future studies are required to collect longitudinal data for the findings. Thus, the findings of this research would be compared with the findings of other research that used to collect data longitudinally. Also, respondents from other regions, such as Russia, can be targeted for data collection because of significant findings in the body of knowledge.

## Supporting information

**S1 File.**
(XLSX)

## Acknowledgments

We acknowledge the efforts of editor and reviewers for their valuable comments and feedback to improve the quality of paper.

## Author Contributions

**Conceptualization:** Saima Kiran.

**Data curation:** Saima Kiran, Faisal Sajjad.

**Formal analysis:** Saima Kiran.

**Funding acquisition:** Saima Kiran.

**Investigation:** Yongming Zhu, Saima Kiran.

**Methodology:** Saima Kiran.

**Resources:** Shahid Sherwani.

**Software:** Muhammad Salman.

**Supervision:** Yongming Zhu, Muhammad Salman.

**Validation:** Yongming Zhu, Muhammad Salman, Naeem Ud Din.

**Visualization:** Yongming Zhu, Muhammad Salman, Shahid Sherwani.

**Writing – original draft:** Muhammad Salman.

**Writing – review & editing:** Muhammad Salman, Faisal Sajjad, Naeem Ud Din.

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
