## [Decision Letter · Decision Letter 0]

4 Sep 2023

PONE-D-23-20140From Crisis to Responsibility: The Role of Industry Type, Leadership Style, and Regulatory Environment in Shaping Post-COVID-19 CSR Initiative.PLOS ONE

Dear Dr. Kiran,

Thank you for submitting your manuscript to PLOS ONE. After careful consideration, we feel that it has merit but does not fully meet PLOS ONE’s publication criteria as it currently stands. Therefore, we invite you to submit a revised version of the manuscript that addresses the points raised during the review process.

We look forward to receiving your revised manuscript.

Kind regards,

Kittisak Jermsittiparsert, Ph.D.

Academic Editor

PLOS ONE

Journal Requirements:

3.PLOS requires an ORCID iD for the corresponding author in Editorial Manager on papers submitted after December 6th, 2016. Please ensure that you have an ORCID iD and that it is validated in Editorial Manager. To do this, go to ‘Update my Information’ (in the upper left-hand corner of the main menu), and click on the Fetch/Validate link next to the ORCID field. This will take you to the ORCID site and allow you to create a new iD or authenticate a pre-existing iD in Editorial Manager. Please see the following video for instructions on linking an ORCID iD to your Editorial Manager account: https://www.youtube.com/watch?v=_xcclfuvtxQ

Reviewers' comments:

Reviewer's Responses to Questions

**Comments to the Author**

1. Is the manuscript technically sound, and do the data support the conclusions?

Reviewer #1: Yes

Reviewer #2: Yes

2. Has the statistical analysis been performed appropriately and rigorously? 

Reviewer #1: Yes

Reviewer #2: Yes

3. Have the authors made all data underlying the findings in their manuscript fully available?

Reviewer #1: Yes

Reviewer #2: Yes

4. Is the manuscript presented in an intelligible fashion and written in standard English?

Reviewer #1: Yes

Reviewer #2: Yes

5. Review Comments to the Author

Reviewer #1: - What novelties are offered?

- An abstract summarizes, usually in one paragraph of 200-259 words or less, the major aspects of the entire paper in a prescribed sequence that includes: 1) the overall purpose of the study and the research problem(s) you investigated; 2) the basic design of the study; 3) major findings or trends found as a result of your analysis; and, 4) a brief summary of your interpretations and conclusions.

- Authors must add citation to these sentences from this article: https://journal.ppishk.org/index.php/jcgpp/article/view/183

- Please add state of the art, gap analysis, and purpose of the paper.

- Methodology is Clear!

- Please add more references from journal articles!

- Please update your references!

Reviewer #2: This article is quite nice and has been produced using acceptable scientific writing procedures.

Minor adjustments are suggested in the conclusion section, with the need to be firm in distinguishing closing points. The ending should be totally revised with the following points in mind:

Brief description of the research topic, copyright gained during the investigation, generalisation of research findings (where each point should be devoted to as an answer provided in the Introduction as mentioned in the Introduction)

6. PLOS authors have the option to publish the peer review history of their article (what does this mean?). If published, this will include your full peer review and any attached files.

Reviewer #1: No

Reviewer #2: **Yes: **

---

## [Author Response · Author response to Decision Letter 0]

26 Sep 2023

Dear Editor and Reviewers,

Reference to the Manuscript PONE-D-23-20140 titled “From Crisis to Responsibility: The Role of Industry Type, Leadership Style, and Regulatory Environment in Shaping Post-COVID-19 CSR Initiative”. The reviewer 1 and review 2 both recommended the publication of this paper with minor revisions. As advised by the worthy referees of Plos one, suggested changes have been incorporated into the manuscript. 

Note: The Red highlighted text indicates the revisions made in the manuscript. We also thoroughly checked and proof read the article and the proof-reading changes are indicated with the blue color.

We hope, the revised manuscript will satisfy the journal’s requirements for publication.

Thanks, and Best Regards,

The Authors

Reviewer #1:

Comment:

What novelties are offered?

Response:

The novelty of research is explained in the introduction section of the article. This research offered novelty based on its findings as it reported CSR initiatives in post-pandemic environment are possible with the type of industry and the style of leadership. These both factors are significant for CSR practices, the regulatory environmental factors are supporting leadership style to lead the work to advance CSR practices.

Comment:

An abstract summarizes, usually in one paragraph of 200-259 words or less, the major aspects of the entire paper in a prescribed sequence that includes: 1) the overall purpose of the study and the research problem(s) you investigated; 2) the basic design of the study; 3) major findings or trends found as a result of your analysis; and, 4) a brief summary of your interpretations and conclusions.

Response:

Dear reviewer, the abstract of the paper is already written according to the standard guidelines. Thanks for your valuable comments.

Comment:

Authors must add citation to these sentences from this article: https://journal.ppishk.org/index.php/jcgpp/article/view/183

Response:

Respected reviewer, the said paper is not fit in the scope of the current scientific research. Therefore, we extend our apology to add this reference. If you have further concerns, please inform us.

Comment:

Please add state of the art, gap analysis, and purpose of the paper.

Response:

Respected reviewer, thanks for your comments. The state-of-the-art gap and purpose of the study is further explained in the introduction section of the study. Please find the highlighted changes.

Comment:

Methodology is Clear!

Response:

Dear reviewer, thanks for your understanding and appreciating our work 

Comment:

Please add more references from journal articles!

Response:

Respected reviewer, the references are updated as per your recommendations. Thanks for your valuable feedback.

Comment:

Please update your references!

Response:

Dear reviewer, thanks for your comments. All references are cited by using end note software. All reference are checked and verified as per the recommendations of worthy reviewer.

Reviewer #2:

Comment:

Minor adjustments are suggested in the conclusion section, with the need to be firm in distinguishing closing points. The ending should be totally revised with the following points in mind:

Brief description of the research topic, copyright gained during the investigation, generalisation of research findings (where each point should be devoted to as an answer provided in the Introduction as mentioned in the Introduction)

Response:

Respected reviewer, thanks for your valuable comments. We are thankful for your critical commentary on this scientific research. In this way, the conclusion section is revised as per your recommendations. Highlighted changes are indicated with red highlighted text.

---

## [Editor Report · Decision Letter 1]

28 Sep 2023

From Crisis to Responsibility: The Role of Industry Type, Leadership Style, and Regulatory Environment in Shaping Post-COVID-19 CSR Initiative.

PONE-D-23-20140R1

Dear Dr. Kiran,

We’re pleased to inform you that your manuscript has been judged scientifically suitable for publication and will be formally accepted for publication once it meets all outstanding technical requirements.

Kind regards,

Kittisak Jermsittiparsert, Ph.D.

Academic Editor

PLOS ONE

Additional Editor Comments (optional):

Thank yo so much for your work and also revision.
---

## [Editor Report · Acceptance letter]

26 Mar 2024

PONE-D-23-20140R1 

PLOS ONE

Dear Dr. Kiran, 

I'm pleased to inform you that your manuscript has been deemed suitable for publication in PLOS ONE. Congratulations! Your manuscript is now being handed over to our production team.

Kind regards, 

on behalf of

Professor Kittisak Jermsittiparsert 

%CORR_ED_EDITOR_ROLE%

PLOS ONE